# One-Pot Synthesis of Al-P-O Catalysts and Their Catalytic Properties for *O*-Methylation of Catechol and Methanol

**DOI:** 10.3390/ma14205942

**Published:** 2021-10-10

**Authors:** Dongfei Xu, Jiaan Ren, Shengnan Yue, Xiujing Zou, Xingfu Shang, Xueguang Wang

**Affiliations:** State Key Laboratory of Advanced Special Steel, Shanghai Key Laboratory of Advanced Ferrometallurgy, School of Materials Science and Engineering, Shanghai University, Shanghai 200444, China; xudongfei@shu.edu.cn (D.X.); 18717902659@163.com (J.R.); snyue@shu.edu.cn (S.Y.); wxg228@shu.edu.cn (X.W.)

**Keywords:** Al-P-O catalysts, P123-assisted, one-pot method, catechol, *O*-methylation

## Abstract

A series of Al-P-O catalysts (Al-*x*P-O) were prepared using a P123-assisted one-pot method at different P/Al molar ratios and used for *O*-methylation of catechol and methanol. The influences of the P/Al molar ratio and P123 addition on catalyst structure and surface acid-base characteristics were investigated in detail. Increasing the P/Al molar ratio more favored crystalline aluminophosphate. The P123-assisted Al^3+^ and PO_4_^3−^ are known to be stabilized through weak steric force so that the formation of crystalline aluminophosphate could be inhibited at higher P/Al molar ratios. The results showed that the prepared Al-P-O catalysts possessed appropriate weak acid and weak base sites, which was beneficial to the reaction of catechol and methanol. The Al-1.1P-O catalyst synthesized with the assistance of P123 exhibited superior catalytic performances, with 52.5% catechol conversion and higher guaiacol selectivity of 97.6%.

## 1. Introduction

Guaiacol is also known as o-hydroxyanisole or o-methoxyphenol. As an important fine chemical raw material and intermediate [1,2,3], guaiacol and its derivatives are widely used in medicine [4,5], agricultural chemic [6,7], and spices [8], especially in synthesis of vanillin [9,10,11,12]. Traditionally, guaiacol is produced mainly by methylation of catechol with halogenated alkanes (CH_3_I, CH_3_CH_2_Br), (CH_3_)_2_SO_4_, or dimethyl carbonate (DMC) in the presence of sodium hydroxide by homogeneous reaction [13,14,15]. In these reaction processes, large amounts of toxic organic compounds required to generate guaiacol would inevitably cause serious environmental concerns. In addition, they involve a lot of secondary reactions and complicated processes, affecting the yield and quality of the products. Therefore, much attention has been paid to developing a green and efficient synthesis process for guaiacol.

Recently, vapour-phase selective *O*-methylation of catechol to synthesize guaiacol, which is an economically and environmentally friendly route for industrial applications, has attracted extensive attention from scientific researchers [16,17,18,19,20]. Methanol is believed to be a very promising candidate for use as a methylation agent for selective *O*-methylation of catechol due to its low toxicity and low cost. The heterogeneous catalysts used for vapour-phase selective *O*-methylation of catechol and methanol is the main factor affecting the yield and quality of the guaiacol. It has been widely believed that solid acid-base catalysts were active catalysts for *O*-methylation of catechol and methanol. Lots of solid acid-base catalysts, such as supported catalysts [13,17,21], molecular sieve-based catalysts [22], phosphate catalysts [16,23,24,25,26,27], Mg-Al hydrotalcites catalysts [18], and oxide or mixed oxide catalysts [14,28,29,30] exhibit catalytic performance for vapour-phase selective *O*-methylation of catechol and methanol. Among various solid acid-base catalysts, phosphate catalysts, especially aluminophosphates, have been widely used for vapour-phase selective *O*-methylation of catechol and methanol because of their high activity and low cost.

Liao et al. [22] found that coating microporous aluminophosphates on mesoporous SBA-15 bearing suitable weak acid-base sites exhibits the relatively high activity of 68.3% and guaiacol selectivity of 90.6%. Zhu et al. [26] found that aluminophosphates with lower Ti content prepared by non-uniform precipitation methods favored guaiacol formation and catalytic stability. During 450-h reaction, the conversion of catechol is above 84%, and the guaiacol selectivity is around 90%. Liu et al. [27] developed a citric acid route to prepare amorphous mesoporous aluminophosphates with different P/Al ratios. Among the prepared samples, AlP_1.1_O showed the highest performance (88.4% conversion of catechol and 84.1% selectivity of guaiacol). These reported aluminophosphate catalysts usually suffer a serious problem of lower guaiacol selectivity and bad stability, increasing the cost of product purification or limiting the application in actual production. Hence, it is still important to develop a simple approach to fabricate aluminophosphates with higher activity, selectivity, and stability for vapor-phase *O*-methylation of catechol with methanol. It has been reported that the catalytic performance of aluminophosphate catalysts could be significantly improved by optimizing catalyst compositions, improving catalyst preparation processes, using additional ligand, carefully selecting operating conditions, etc. [14,25,26].

In this work, a series of Al-P-O catalysts (Al-*x*P-O) were prepared by a P123-assisted one-pot method at different P/Al molar ratios, which exhibited excellent catalytic activity, guaiacol selectivity, and stability for vapour-phase selective *O*-methylation of catechol and methanol. The effects of the P/Al molar ratio and P123 addition on catalyst structure and surface acid-base characteristics were investigated in detail.

## 2. Materials and Methods

### 2.1. Catalyst Preparation

All chemicals were purchased and used without further purification. Pluronic P123 (EO_20_PO_70_EO_20_) was bought from Sigma-Aldrich Reagent (Shanghai, China). Other chemicals were supplied by Sinopharm Chemical Reagent Co., Ltd. (Shanghai, China).

AlCl_3_ (99.0%) was used as the aluminum source, H_3_PO_4_ (85%) was used as the phosphorus source, and the block copolymer P123 (M_av_ = 5800) was used as the additional ligand. A series of Al-P-O catalysts with different P/Al molar ratios were prepared using a P123-assisted one-pot method. Typically, 44.5 g AlCl_3_ and 16.9 g P123 were firstly dissolved in 200 mL of deionized water respectively. After that, the aqueous solution of AlCl_3_ and P123 were directly mixed with vigorous magnetic stirring and heated to 60 °C in a water bath. Then, a required H_3_PO_4_ was poured into the mixed aqueous solution. Following this, 6 mol/L ammonium hydroxide solution were dropped into the above solution until pH value of the mixture was ca. 9. The mixture was further stirred for 1 h. Then, the mixture was evaporated at 80 °C, dried at 100 °C for 12 h, and calcined with a heating rate of 2 °C/min in air at 550 °C for 12 h in sequence. The prepared materials were labelled as Al-*x*P-O, where *x* represented Al/P molar ratio.

For comparison, Al-1.1P-O^a^ was also prepared by the same approach as Al-1.1P-O without P123 addition.

### 2.2. Catalyst Characterization

Powder X-ray diffraction (XRD) measurements of the samples were performed with a Germany Bruker D8 Advance ray diffractometer using Cu Kα radiation at 40 kV and 40 mA. The diffraction angle (2*θ*) ranged from 10° to 70°, and the scanning speed was 8°/min. The crystallite size of AlPO_4_ was calculated using the full-widths at half maximum (FWHM) of the AlPO_4_ (0 2 0) peak through the Scherrer equation.

N_2_ adsorption-desorption isotherms were performed on the Micromeritics ASAP 2020 sorptometer at liquid nitrogen temperature (−196 °C). Before the measurement, each catalyst was degassed at 250 °C for 5 h. The specific surface area (*S*_BET_) was calculated by the Brunauer-Emmett-Teller (BET) method within the range of relative pressure *P*/*P*_0_ = 0.10~0.30. The average pore size (*D*_a_) was calculated using the BJH method. The pore volume (*V*_p_) was set as the single point value when *P*/*P*_0_ was 0.990.

SEM images were acquired by employing a FEI Nova nanoSEM 450 electron microscope. The samples were sprayed with metal layer before testing.

FT-IR spectra were performed on a Germany Bruker TENSOR 27 Fourier transform infrared spectrometer. The structure of the sample and the vibration of the skeleton were made using a KBr support chip. (The mass ratio of catalyst to KBr is 1:100).

Temperature programmed desorption (TPD) was performed on a Biode PCA-1200 chemisorption analyzer for the surface acid (base) center strength and number of the catalysts by using NH_3_ (CO_2_) as the probe molecule and TCD as the detector. Prior to the measurement, a 100 mg sample was accurately weighed and placed in a quartz tube and then heated to 400 °C at a rate of 10 °C/min under Ar flow (30 mL/min) for 1 h to remove the moisture in samples and the residual gas in the test system. Then it was cooled to 50 °C and exposed to NH_3_ (CO_2_) for 0.5 h. Ar flow (30 mL/min) was purged from the quartz tube to remove the physical adsorption of NH_3_ (CO_2_) from the sample surface. Finally, the sample was heated at a rate of 10 °C/min to the specified temperature in Ar flow (30 mL/min), and the desorption temperature of NH_3_ (CO_2_) was detected using a thermal conductivity detector (TCD). The total surface acidity and basicity of the samples were determined from the TPD of NH_3_ and CO_2_, respectively.

The TGA of the catalysts was investigated on a Netzsch STA 4449 F3, which was used to measure the carbon deposition of the catalyst after reaction.

### 2.3. Catalytic Performance Evaluation

Vapour-phase selective *O*-methylation synthesis of guaiacol by catechol and methanol was performed in a self-made fixed-bed reactor at atmospheric pressure with a length of 750 mm and an inside diameter of 14 mm quartz tube. The catalyst evaluation system was shown in Figure 1. A 6 g catalyst (20–40 mesh) was taken and placed in a catalytic bed. Prior to the reaction, nitrogen was used to remove oxygen from the tube, the catalyst was heated at a rate of 10 °C/min in the meantime, and nitrogen purging was stopped until the catalyst was heated to a specified reaction temperature. Reaction liquid was transported into the reaction tube by advection pump. The products were analyzed with a gas chromatograph (Fuli gas chromatograph GC9790) equipped with a capillary column (SE-54) and were identified using known standards and GC–MS. Guaiacol is the main product and 1,2-dimethoxybenzene is the main byproduct. Concurrently, other byproducts are mainly further alkylates of the products as follows:

The conversion of catechol (*C*_catechol_), selectivity of guaiacol (*S*_guaiacol_), selectivity of 1,2-dimethoxybenzene (*S*_1,2-dimethoxybenzene_), selectivity of other byproducts (*S*_other byproducts_), and yield of guaiacol (*Y*_catechol_) were calculated using the normalization method, with catechol as the reference substance. The details are shown in Equations (1)–(5):(1)$$C_{\mathrm{catechol}}=\frac{f_{1}A_{1}+f_{2}A_{2}+f_{3}A_{3}}{f_{1}A_{1}+f_{2}A_{2}+f_{3}A_{3}+A}\times 100\%$$
(2)$$S_{\mathrm{guaiacol}}=\frac{f_{1}A_{1}}{f_{1}A_{1}+f_{2}A_{2}+f_{3}A_{3}}\times 100\%$$
(3)$$Y_{\mathrm{guaiacol}}=C_{\mathrm{catechol}}\times S_{\mathrm{guaiacol}}÷100$$
(4)$$S_{1,2-\mathrm{Dimethoxybenzene}}=\frac{f_{2}A_{2}}{f_{1}A_{1}+f_{2}A_{2}+f_{3}A_{3}}\times 100\%$$
(5)$$S_{\mathrm{other}\;\mathrm{byproducts}}=\frac{f_{3}A_{3}}{f_{1}A_{1}+f_{2}A_{2}+f_{3}A_{3}}\times 100\%$$
where *f*_1_, *f*_2_, and *f*_3_ represented the correction factor of guaiacol; 1,2-dimethoxybenzene; and other byproducts, respectively. In view of the similar properties and low content of these byproducts, we calculate the other byproducts as a whole. The calibration factor of 2-methoxy-6-methylphenol is used to calculate other byproducts, where *A*, *A*_1_, *A*_2_, and *A*_3_ represent chromatographic peak areas of catechol; guaiacol; 1,2-dimethoxybenzene; and the sum of peak areas of all byproducts.

## 3. Results and Discussion

### 3.1. Catalyst Characterization

Figure 2 presents the XRD patterns of Al-*x*P-O (*x* = 0, 0.25, 0.05, 0.75, 1.00, 1.10, 1.15, 1.20) catalysts, together with that of Al-1.1P-O^a^ for comparison. The Al-0P-O sample showed three strong diffraction peaks of γ-Al_2_O_3_ around 2*θ* = 37°, 46°, and 67°. When the P/Al molar ratio increased to 0.25, it was interesting to note that the diffraction peaks intensities of γ-Al_2_O_3_ decreased significantly, and a new weak and broad diffraction peak corresponding to amorphous aluminum phosphate around 2*θ* = 24° appeared [22,30,31]. No diffraction peaks assigned to P_2_O_5_ were observed. These results demonstrated that the addition of P resulted in forming amorphous aluminum phosphate rather than a mixture of Al_2_O_3_ and P_2_O_5_. With the increasing of the P/Al molar ratio to 0.5, the diffraction peak around 2*θ* = 24° was observed clearly. Concurrently, the diffraction peaks corresponding to γ-Al_2_O_3_ completely disappeared. With further increase of the P/Al molar ratio to 1.1, the intensity of aluminum phosphate diffraction peaks kept no obvious change. When the P/Al molar ratio further increased to 1.15, four sharp diffraction peaks assigned to aluminum phosphate around 2*θ* = 20.4°, 21.6°, 23.2°, and 35.8° were observed, corresponding to lattice planes of (0 2 0), (2 1 1), and (2 1 2) of tridymite and a lattice plane of (2 6 0) of α-cristobalite, respectively [32]. In addition, diffraction peaks of NH_4_AlP_2_O_7_ were also noticed around 2*θ* = 29.8°, 31.1°, and 38.7°. The intensity of AlPO_4_ crystalline phase and NH_4_AlP_2_O_7_ crystalline phase became stronger in the Al-1.20P-O sample than in Al-1.15P-O. Table 1 displays the crystallite size of AlPO_4_ calculated using the Scherrer equation. AlPO_4_ crystallite sizes were 30.9 nm for the Al-1.15P-O and 34.3 nm for the Al-1.20P-O, respectively. These results suggest that AlPO_4_ crystalline phase and NH_4_AlP_2_O_7_ crystalline phase formed when the P/Al molar ratio reached 1.15. Larger crystalline sizes are easier to form on Al-*x*P-O with higher P/Al molar ratios. Compared with the Al-1.1P-O, the Al-1.1P-O^a^ sample showed distinct crystalline phases of AlPO_4_ and NH_4_AlP_2_O_7_. These results demonstrated that the presence of P123 in the precursor solutions could restrain the formation of crystalline phases in regard to Al and P species.

The basic textural properties of the prepared Al-P-O samples are summarized in Table 1. It could be seen that compared with Al-0P-O, the prepared Al-0.25P-O sample exhibited an increase in specific surface area, pore volume, and average pore size. When the P/Al molar ratio increased from 0.25 to 1.10, a decrease in the surface area and pore volume was observed, but the pore sizes had little change. In further increasing the P/Al molar ratio to 1.15 and 1.20, the specific surface area, pore volume and average pore size decreased dramatically. For instance, the Al-0.25P-O sample showed a specific surface area (*S*_BET_) of 319 m^2^ g^−1^, a pore volume (*V*_P_) of 2.03 cm^3^ g^−1^, and an average pore size (*D*_a_) of 21.6 nm, whereas those of the Al-1.10P-O were 147 m^2^ g^−1^, 1.08 cm^3^ g^−1^, and 23.9 nm, respectively. However, Al-1.15P-O only had a *S*_BET_ of 30 m^2^ g^−1^, *V*_P_ of 0.10 cm^3^ g^−1^ and *D*_a_ of 14.2 nm. Compared with the Al-1.1P-O sample, the Al-1.1P-O^a^ sample-prepared absence of P123 showed smaller specific surface area, pore volume, and average pore size. Combined with XRD results, it can be speculated that P addition can bond with Al^3+^ to form aluminum phosphate, further altering the basic textural properties of the Al-*x*P-O samples. Higher P/Al molar ratios decreased specific surface area, pore volume, and average pore size. The existence of P123 favors a higher specific surface area, pore volume, and average pore size.

The microscopic morphology of Al-*x*P-O samples are showed in Figure 3. It can be clearly observed that while the P/Al ratio was less than or equal to 1.1, the samples had a wormlike structure, and the microscopic morphology of samples did not change significantly with the increase of the P/Al ratio. While the P/Al ratio was higher than 1.1, the particles of different sizes were deposited on the surface of the sample to form the accumulation holes, and the grain size increased prominently with the increase of the P/Al molar ratio. These indicated that the samples with a higher P/Al ratio (*x* > 1.1) were easy to condense to particles with crystal phase structure, leading to a prominent decrease in the specific surface area of samples, which was consistent with XRD and BET results. As for the Al-1.1P-O^a^ sample, a dense and less porous morphology was observed.

FT-IR spectra of Al-*x*P-O samples are presented in Figure 4. All the tested samples had obvious vibration signals at 3470 cm^−1^ and 1634 cm^−1^, which were attributed to the stretching vibration the O-H bond and bending vibration of physically adsorbed water, respectively [24,33,34,35]. All samples except Al-0P-O had vibration signals at 1139cm^−1^, 730 cm^−1^ and 490 cm^−1^. The vibration signal at 1139 cm^−1^ could belong to the asymmetric stretching vibration of the P-O bond in the (PO_4_)^3−^ tetrahedron. The vibration signal located around 490 cm^−1^ was attributed to the bending vibration of O-P-O in the (PO_4_)^3−^ tetrahedron. The vibration signal around 730 cm^−1^ was assigned to the vibration of Al-O bond combined with P-O bond [27,31,36,37]. When *x* ≤ 1.1, Al-*x*P-O samples had a weak vibration signal near 730 cm^−1^, this indicated that Al-*x*P-O samples (*x* < 1.1) had an amorphous aluminum phosphate structure instead of a simple mixture of Al_2_O_3_ and P_2_O_5_. When *x* > 1.1, the vibration signal of the Al-*x*P-O samples at 730 cm^−1^ was significantly enhanced. This strong vibration signal should be caused by the formation of crystalline aluminum phosphate phase, which is consistent with the results of XRD characterization.

NH_3_-TPD profiles and surface acidic and basic parameters of Al-*x*P-O and Al-1.1P-O^a^ samples are displayed in Figure 5a and Table 1. All samples had a desorption peak between 100 and 250 °C, indicating the existence of weak acid sites [38]. The desorption peaks shifted to lower NH_3_ desorption temperature with the increasing P/Al molar ratio, indicating that the addition of P weakened the strength of the acid site. The area of the NH_3_ desorption peak increased when the P/Al molar ratio increased from 0 to 0.75. With the increase of the P/Al molar ratio to 1.10, the area of the NH_3_ desorption peak decreased slightly. However, when the P/Al molar ratio further increased to 1.15 and 1.2, a sudden decrease of the area of the NH_3_ desorption peak appeared, indicating that the number of acid sites of catalysts with higher P/Al molar ratio (*x* ≥ 1.15) was very low. It could be speculated that the P-OH group was the main source of the weak acid center. The catalyst with the higher P/Al molar ratio (P/Al > 1.1) eased the formation of aluminum phosphate crystal phase, resulting in a low number of acid centers. Compared with the Al-1.1P-O sample, Al-1.1P-O^a^ had a lower NH_3_ desorption temperature and a smaller area of NH_3_ desorption peak, proving that P123 addition could boost the formation of stronger acid sites.

CO_2_-TPD profiles of Al-*x*P-O samples are shown in Figure 5b. All the samples gave a broad CO_2_ desorption temperature in the range of 100–350 °C. The desorption peak in the ranges of 100–200 °C and 200–350 °C were assigned to weak basic site and medium-strength basic site [27]. The CO_2_ desorption peaks shifted towards low temperature with the increasing P/Al molar ratio, indicating that the strength of the basic sites gradually decreased. However, the area of the CO_2_ desorption peak for the catalysts with a P/Al molar ratio between 0.25 and 1.1 showed no obvious changes, whereas the area of the CO_2_ desorption peak decreased significantly in Al-*x*P-O samples (P/Al > 1.1). In the case of Al-1.1P-O^a^, a lower CO_2_ desorption temperature and a smaller area of the CO_2_ desorption peak than those of the Al-1.1P-O sample indicate that P123 addition could promote the formation of stronger basic sites.

### 3.2. Catalytic Performance Evaluation

The catalytic performance of Al-*x*P-O catalysts for vapour-phase selective *O*-methylation of catechol and methanol to guaiacol was investigated under optimized reaction conditions, and the results are listed in Table 2. The P/Al molar ratio had a significant influence on the catalyst activity and selectivity. The Al-0P-O showed catechol conversion of 45.1%. With the addition of P, the catechol conversion gradually increased and showed a maximum value of 76.8% at *x* = 0.75. As the P/Al molar ratio increased further, the catalytic activity started to decrease, and the catechol conversion decreased to 58.0% at *x* = 1.10, further increasing the P/Al molar ratio to *x* = 1.15, only a 25.2% conversion. These results were in accordance with the number of acid sites measured by NH_3_-TPD shown in Figure 5. The low catechol conversion was mainly due to scarcity of acidity sites. Guaiacol selectivity firstly increased with the increasing P/Al molar ratio in the range of 0–1.1, and then increased slightly at 97.6% and 99.4% for Al-1.1P-O and Al-1.2P-O. Comparatively, Al-1.1P-O^a^ catalysts without P123 showed lower catechol conversion of 43.6% compared to Al-1.1P-O (catechol conversion of 58.0%) but also showed a high selectivity of 97.7%. Combined with the characterization results, the Al-*x*P-O catalysts which possessed weak acid-base sites exhibited excellent guaiacol selectivity with higher catalytic activity. Although the Al-0.75P-O catalyst had the highest yield, it was worth noting that multiple byproducts were generated, including 3-methylcatechol; 2-methoxy-6-methylphenol; 2,3-dimethoxytoluene; etc. Poor selectivity certainly would increase the cost of product separation. Therefore, Al-1.10P-O catalysts, which exhibited a higher selectivity of 97.6% with a comparatively higher conversion of 58.0%, was considered to be chosen for further stability study.

The catalytic stability test was conducted on the Al-1.1P-O catalyst under the reaction conditions m_cat_ = 6 g, Reaction temperature = 275 °C, LHSV = 0.6 h^−1^, methanol/catechol = 6 mol. To understand the effect of P123 on the stability of the catalyst, a stability test was also carried out on an Al-1.10P-O^a^ sample under the same conditions. As can be seen in Figure 6, the addition of P123 had a significant influence on catalytic activity and selectivity. For the Al-1.1P-O catalyst, guaiacol selectivity was constant at ca. 97.6% during the 300-h reaction period. The conversion of catechol decreased slightly from ca. 58.0% to ca. 53.0% in the initial stage of 50 h, and kept unchanged during the following period, showing very high catalytic stability. In the case of Al-1.10P-O^a^, catechol conversion and guaiacol selectivity decrease from ca. 43.6% and ca. 97.7% to ca. 18.0% and ca. 95.9%, respectively, during the 100-h reaction time.

It has been established that carbon deposition is mainly responsible for the deactivation of acid/base catalysts. Thus, TG was employed to investigate the amount of carbon depositions on the spent Al-1.1P-O catalyst after 300 h (denoted as Al-1.1P-O-S300) and the spent Al-1.10P-O^a^ catalyst (denoted as Al-1.1P-O^a^-S100) after 100 h for *O*-methylation of catechol. The samples exhibited a first weight loss in the low temperature range of ca. 50–250 °C, corresponding to physically adsorbed commands. This process was followed by the removal of the carbon deposited on the catalyst surface in the range of ca. 250–650 °C, which was used to quantify the amount of deposited carbon. When the temperature reached 650 °C, negligible weight loss was detected. The results of the amounts of carbon depositions are given in Table 3. Only trace amounts of carbon were deposited in the Al-1.1P-O catalyst. The average carbon depositions per hour on Al-1.1P-O^a^-S100 were three times as high as those on Al-1.1P-O, indicating that the addition of P123 was indeed beneficial to inhibit and eliminate the carbon deposition. Combined with the catalyst characterization results, we speculated that smaller pore volume suppressed the large molecular size byproducts detaching from the catalyst surface. As the reaction went on, the active center of the reaction was gradually occupied, leading to the gradual decline of catechol conversion.

## 4. Conclusions

In summary, Al-*x*P-O samples were prepared through a P123-assisted one-pot method and used for *O*-methylation of catechol and methanol to produce guaiacol. Al and P species were mainly in the aluminum phosphate phase. When *x* ≤ 1.1, the aluminum phosphate phase was amorphous, and the aluminum phosphate in crystalline phase with large particle size formed in Al-*x*P-O samples with higher P/Al molar ratios (*x* > 1.1). P123 addition could inhibit the formation of aluminum phosphate crystal phase to a certain extent. P/Al molar ratio and P123 addition had a significant influence on crystal structure, acid-base sites, and basic textural properties. All prepared samples showed weak acid-base sites, which were beneficial to producing guaiacol with catechol and methanol. The Al-1.1P-O catalyst prepared with P123-assistance exhibited superior catalytic performances with ca. 58.0% catechol conversion, ca. 97.6% guaiacol selectivity, and 300 h stability.

## Figures and Tables

**Figure 1 materials-14-05942-f001:**
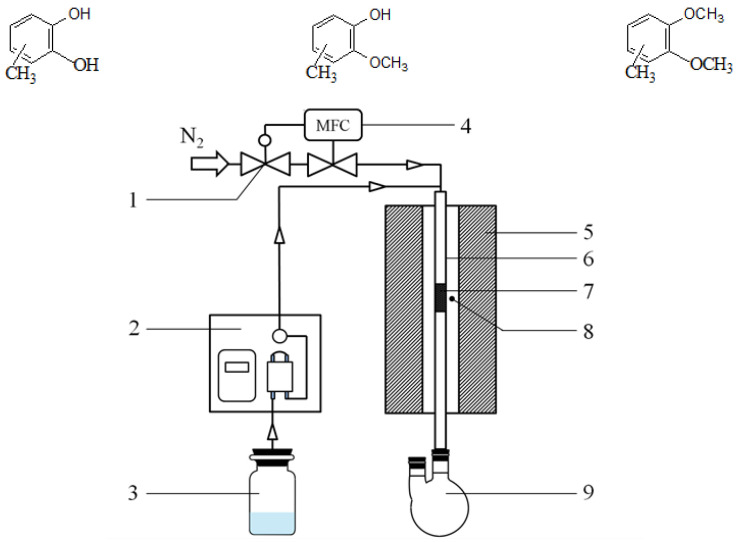
Diagram of catalyst evaluation system. 1. Stable-Flow valve; 2. Advection pump; 3. Reaction mixture; 4. Mass flow controller; 5. Tube furnace; 6. Reaction tube; 7. Catalyst bed; 8. Thermoelectric couple; 9. Receiving flask.

**Figure 2 materials-14-05942-f002:**
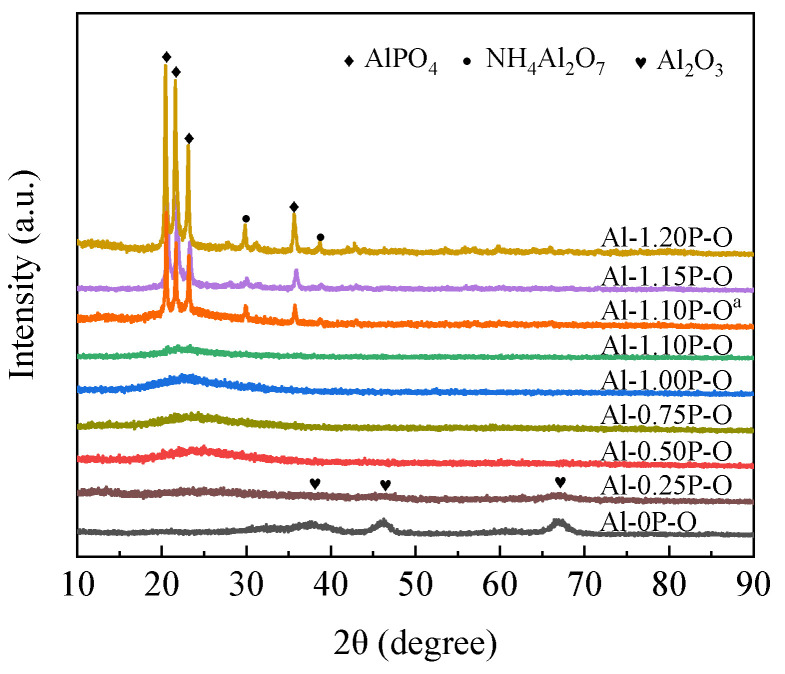
XRD patterns of Al-*x*P-O and Al-1.1P-O^a^ samples.

**Figure 3 materials-14-05942-f003:**
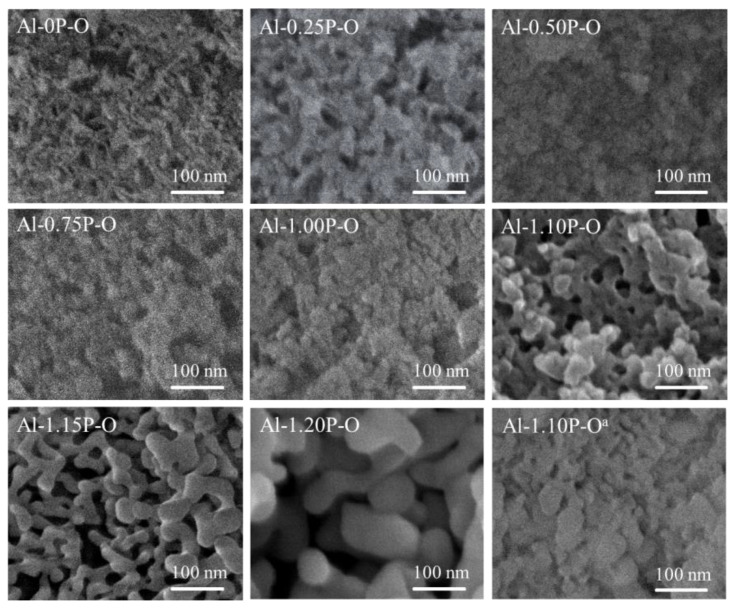
SEM images of Al-*x*P-O and Al-1.1P-O^a^ samples.

**Figure 4 materials-14-05942-f004:**
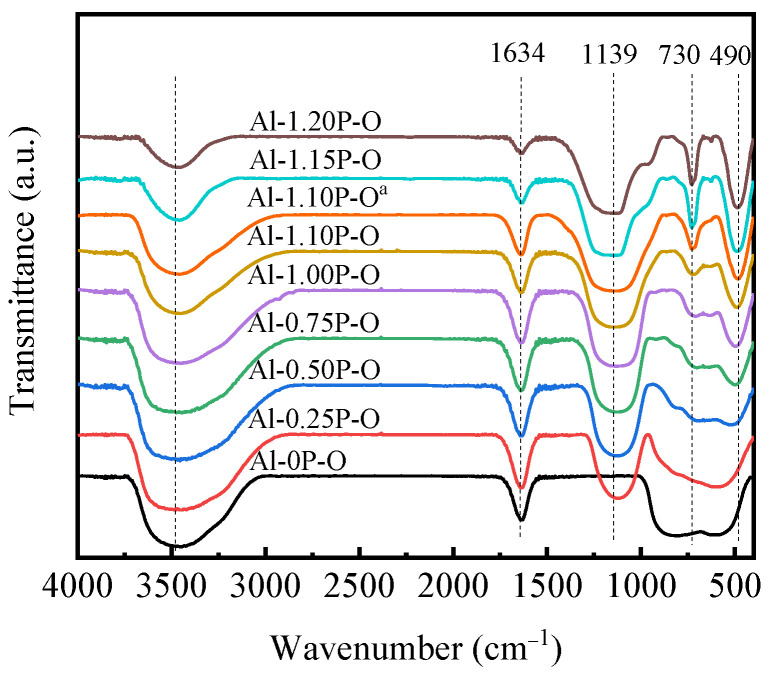
FT-IR spectra of Al-*x*P-O and Al-1.1P-O^a^ samples.

**Figure 5 materials-14-05942-f005:**
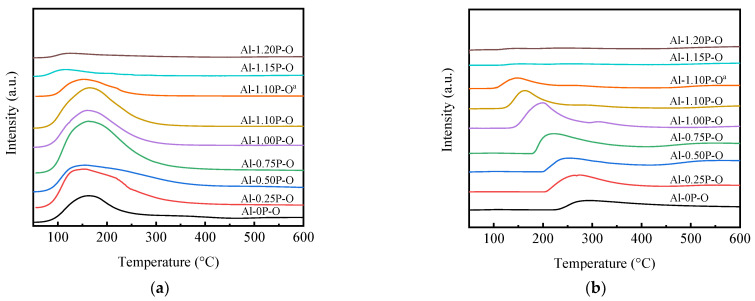
(**a**) NH_3_-TPD and (**b**) CO_2_-TPD profiles of Al-*x*P-O and Al-1.1P-O^a^ samples.

**Figure 6 materials-14-05942-f006:**
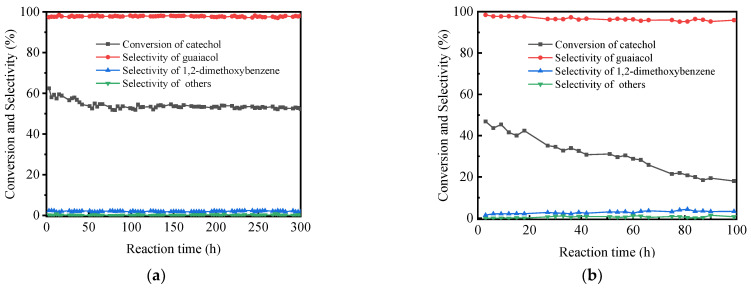
Stability test of (**a**) Al-1.1P-O catalyst and (**b**) Al-1.1P-O^a^. Reaction conditions: m_cat_ = 6 g, Reaction temperature = 275 °C, LHSV = 0.6 h^−1^, methanol/catechol = 6 mol.

**Table 1 materials-14-05942-t001:** The textural properties of Al-*x*P-O and Al-1.1P-O^a^ samples.

Catalysts	AlPO_4_ Particle Size (nm) by XRD	S_BET_ (m^2^·g^−1^)	V_P_ (cm^3^·g^−1^)	D_a_ (nm)	Total Acidity (μmol·g^−1^)	Total Basicity (μmol·g^−1^)
Al-0P-O	─	249	1.19	14.9	27.14	34.06
Al-0.25P-O	─	319	2.03	21.6	40.45	32.99
Al-0.50P-O	─	273	1.72	23.7	56.16	28.75
Al-0.75P-O	─	220	1.40	22.6	82.63	46.63
Al-1.00P-O	─	176	1.25	22.9	56.65	35.70
Al-1.10P-O	─	147	1.08	23.9	53.44	21.61
Al-1.10P-O^a^	37.8	99	0.47	15.2	21.96	13.17
Al-1.15P-O	30.9	30	0.10	14.2	5.76	4.15
Al-1.20P-O	34.3	23	0.05	10.7	4.20	3.79

**Table 2 materials-14-05942-t002:** Catalytic performance of Al-*x*P-O and Al-1.1P-O^a^ catalysts.

Catalysts	Conversion of Catechol (%)	Selectivity (%)			Yield (%)
		Guaiacol	1,2-Dimethoxybenzene	Others	
Al-0P-O	45.1	77.4	2.1	20.5	34.9
Al-0.25P-O	64.1	83.6	4.4	12.0	53.6
Al-0.50P-O	66.5	83.1	5.0	11.9	55.3
Al-0.75P-O	76.8	89.5	3.1	7.4	68.7
Al-1.00P-O	63.1	92.4	2.1	5.5	58.3
Al-1.10P-O	58.0	97.6	2.3	0.1	56.6
Al-1.10P-O^a^	43.6	97.7	2.2	0.1	42.6
Al-1.15P-O	25.2	99.3	0.7	0	25.0
Al-1.20P-O	22.9	99.4	0.6	0	22.8

Reaction conditions: m_cat_ = 6 g, Reaction temperature = 275 °C, LHSV = 0.6 h^−1^, methanol/catechol = 6 mol, reaction time = 6 h.

**Table 3 materials-14-05942-t003:** The textural properties of Al-*x*P-O and Al-1.1P-O^a^ samples.

Catalysts	Reaction Time (h)	Carbon Amount by TG (mg·g_cat_^−1^)	Average Carbon Depositions per Hour (mg·g_cat_^−1^·h^−1^)
Al-1.1P-O-S300	300	38	0.13
Al-1.1P-O^a^-S100	100	45	0.45

## Data Availability

The data presented in this study are available on request from the corresponding author.

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
