# Peer review of "One-Pot Synthesis of Al-P-O Catalysts and Their Catalytic Properties for O-Methylation of Catechol and Methanol"

_materials, 2021, doi:10.3390/ma14205942_

Round 1
Reviewer 1 Report
The manuscript by Shang et. al. deals with the catalytic conversion of catechol into guaiacol employing a series of Al-P-O catalysts. Although the interest of the subject, the manuscript lacks of novelty since superior results on this transformation was achieved by other authors. Also, the manuscript is not well written, making it very difficult to the readers to follow the author’s approach. Some specific points are described below:
1 – More experimental details should be provided. For example, how was performed the catechol and other products quantification? (Calibration curve, standard addition?). The chromatograms for all reactions should be provide as supporting information. Also, it is not clear the how the authors found the number of moles of byproducts.
1 – The quality of the Figures should be enhanced. Fig 1(b) the peaks are overlapped (same for Fig. 4).
2 – The XRD discussion is confuse. The authors should be clear on the difference of Fig. 2(a) and Fig.2(b). Also, there is not a reasonable explanation regarding to the difference of the XRD patterns between from (a) to (b).
3 – The authors should quantify the weak and strong acid and basic sites, correlated with the surface area. Thus, correlate with the catalytic activity. In the present form, the discussion of surface area and acidity and basicity of the catalysts is useless.
4 – The argumentation on choosing Al1.1P-O is weak. Al0.75 gave more than 10% of yield and the selectivity for guaiacol (89.5%) is not poor. Also, the authors did not raise any discussion about the by-products to support the claim “it had plenty of by-products and poor selectivity, which increased the cost of product separation”. A list of the main by-products identified should be provide for all catalysts. Also, the authors should provide, as supporting information, the same curves presented in Fig. 6 for all catalysts.
5- There are examples in literature that employs TGA to quantify and identify the nature of carbon deposited in the catalyst. Further discussion on this matter should be added.
6 – The authors should provide a comparison with previous literature data regarding to catechol conversion over aluminum-phosphate catalysts and clearly indicate what is the advance achieved by the present manuscript.
On the present form, the manuscript is not acceptable for publication. The authors must answer, in details, the points raised and a new evaluation will be required for a final decision.
Reviewer 2 Report
The manuscript "One-pot synthesis of Al-P-O catalysts and their catalytic properties for O-methylation of catechol and methanol" describes the preparation and the use of a new kind of catalysts made by Al-P-O for the O-methylation of cathecol and methanol.
The manuscript is clear and the conclusions are supported by the results; however, several points should be revised and therefore I suggest to publish it only after major revisions.
1) The abstract is too coincise: the state of art and P123 method should be better explained.
2) materials: data about suppliers, purity and so on of materials used are missing.
3) the authors should explain the possible reason of low catechol conversion (Table 2) and how it is possible to enhance conversion and yield.
Round 2
Reviewer 1 Report
After the revision by the authors, although the GC-MS total ion chromatograms were not provided, the manuscript is accepted.
Reviewer 2 Report
All my comments have been taken into consideration and therefore I suggest to consider this version of the manuscript suitable for a MDPI publication